# Health Benefits of Physical Activity Related to an Urban Riverside Regeneration

**DOI:** 10.3390/ijerph16030462

**Published:** 2019-02-05

**Authors:** Cristina Vert, Mark Nieuwenhuijsen, Mireia Gascon, James Grellier, Lora E. Fleming, Mathew P. White, David Rojas-Rueda

**Affiliations:** 1ISGlobal (Global Health Institute Barcelona), 08003 Barcelona, Spain; mark.nieuwenhuijsen@isglobal.org (M.N.); mireia.gascon@isglobal.org (M.G.); 2Department of Experimental and Health Sciences, Universitat Pompeu Fabra (UPF), 08002 Barcelona, Spain; 3CIBER Epidemiología y Salud Pública (CIBERESP), 28029 Madrid, Spain; 4European Centre for Environment and Health, University of Exeter College of Medicine and Health, Truro, Cornwall TR1 3HD, UK; J.Grellier@exeter.ac.uk (J.G.); L.E.Fleming@exeter.ac.uk (L.E.F.); Mathew.White@exeter.ac.uk (M.P.W.); 5Department of Epidemiology and Biostatistics, Imperial College, London SW7 2AZ, UK

**Keywords:** urban regeneration, urban health, blue spaces, physical activity, health impacts

## Abstract

The promotion of physical activity through better urban design is one pathway by which health and well-being improvements can be achieved. This study aimed to quantify health and health-related economic impacts associated with physical activity in an urban riverside park regeneration project in Barcelona, Spain. We used data from Barcelona local authorities and meta-analysis assessing physical activity and health outcomes to develop and apply the “Blue Active Tool”. We estimated park user health impacts in terms of all-cause mortality, morbidity (ischemic heart disease; ischemic stroke; type 2 diabetes; cancers of the colon and breast; and dementia), disability-adjusted life years (DALYs) and health-related economic impacts. We estimated that 5753 adult users visited the riverside park daily and performed different types of physical activity (walking for leisure or to/from work, cycling, and running). Related to the physical activity conducted on the riverside park, we estimated an annual reduction of 7.3 deaths (95% CI: 5.4; 10.2), and 6.2 cases of diseases (95% CI: 2.0; 11.6). This corresponds to 11.9 DALYs (95% CI: 3.4; 20.5) and an annual health-economic impact of 23.4 million euros (95% CI: 17.2 million; 32.8 million). The urban regeneration intervention of this riverside park provides health and health-related economic benefits to the population using the infrastructure.

## 1. Introduction

Natural outdoor environments in cities have positive impacts on health and well-being [1]. Living in a green environment has been positively related to better general health [2], self-perceived general health and better mental health [3], reduced perceived stress [4], slower cognitive decline [5], and reduced mortality [6,7]. Visits to green spaces have been found to increase mental well-being [8,9,10]. Some studies suggest that higher levels of greenness are associated with higher levels of physical activity [11].

The health impacts of blue spaces, defined in the European Commission H2020-funded project BlueHealth (https://bluehealth2020.eu/) as “outdoor environments—either natural or manmade—that prominently feature water and are accessible to humans either proximally (being in, on, or near water) or distally/virtually (being able to see, hear or otherwise sense water)” [12], have been studied less than those related to green spaces [13]. A recent systematic review suggests that exposure to outdoor blue spaces is positively associated with improved mental health and well-being, and promotes physical activity [13]. For example, living nearer to the coast is associated with increased numbers of people achieving physical activity guidelines [14,15,16], which may in turn lead to better general and mental health [17].

Physical activity in natural environments is associated with increased well-being compared to physical activity in built environments [18,19]; and efforts to quantify the benefits of physical activity in these natural environments, in terms of welfare gains and lives saved, have begun to emerge [20]. These results are important, given that physical inactivity is the fourth-leading risk factor for mortality worldwide; and it has major implications for the prevalence of non-communicable diseases and general health [21]. Regular physical activity may prevent overweight and obesity, cardiovascular disease, type 2 diabetes, some cancers, and psychological disorders [16,22].

The world is increasingly urbanized: 50% of the world population was living in urban settlements in 2016, and this is expected to rise to 60% by 2030 [23]. Moreover, a third of all people will live in large cities with at least half a million inhabitants by this time [23]. Considering that some aspects of cities—such as scarcity of natural environment—can negatively affect human health, the incorporation of natural (blue and green) outdoor environments in urban planning is a fundamental characteristic of a healthy city [24]. With an adequate urban design, healthy natural and built environments can be achieved in cities [25]; and these may promote healthier behaviors and lifestyles, therefore reducing health-related costs as well as other co-benefits [26]. Such co-benefits may play an important role in reducing the disease burden associated with aspects of urban living such as air pollution, noise, and lack of natural spaces where people can engage in health-promoting physical activities, sedentary behavior, obesity, poor mental health, and other non-communicable chronic diseases [27,28,29,30]. The regeneration of under-used, inadequately designed, or decayed urban spaces (including natural outdoor environments located in urban areas) is now a relatively common phenomenon globally, but not many studies have estimated the impacts of existing interventions in terms of health and well-being [31,32,33,34,35,36,37].

This study focuses on an urban regeneration project on the Besòs River, located in the northeast of Barcelona (Catalonia, Spain). The final stages of the Besòs River flow through an industrialized area prior to entering the Mediterranean Sea (Figure 1a). Since the mid-1990s, the river and its surroundings have undergone considerable infrastructural improvements through urban riverside regeneration. A nine-kilometer long stretch of recreation area—the *Parc Fluvial del Besòs* (Besòs Riverside Park)—was created on the banks of the river, spanning the cities of Barcelona (1.6 million inhabitants), Santa Coloma de Gramenet (117,153 inhabitants), and Sant Adrià de Besòs (36,496 inhabitants) (Figure 1a) [38]. The first stage of this urban riverside regeneration was completed in 2000, and included environmental remediation and the development of a green area on the riverbanks (i.e., 22 hectares mostly covered by grass, being one of the most important green areas of the Barcelona metropolitan area nowadays), as well as the provision of paths for walking and cycling along the river [39] (Figure 1b). The urban riverside park regeneration project was developed to improve the environmental conditions of the area—mainly in terms of its ecological state—as well as to provide spaces for leisure and physical activity and to facilitate social interactions. Prior to the project’s completion, the riverbanks were not accessible and the public was not able to use the area. The aim of this study was to quantify potential health and health-related economic impacts derived from the physical activity related to this new urban riverside park.

## 2. Methods

### 2.1. Input Data

For this study, we used data collected in 2014–2015 by Barcelona local authorities to characterize park users [41]. Two different data sets were used: (1) surveys administered to the park users (*N* = 973) to characterize the main activities performed in the park; and (2) a counting campaign (manual and automated) to estimate the total number of users per year [specifying whether they were cyclists (*N* = 1,030,000) or pedestrians (*N* = 1,070,000)] in the park. The survey was administered to users in the park between 8 am and 8 pm both on weekdays and weekends, in three different points in time: November 2014, July 2015, and September 2015, in order to take seasonality into account. The survey included questions about user characteristics (sex, age, and city of residence), the main activity performed in the park (e.g., walking, cycling, running, etc.), the duration (hours/day), and the frequency of their visits (days/week) (Appendix A).

In this study, we focused on adult respondents (≥18 years old) because the majority of the quantitative epidemiological evidence on associations between physical activity and health is derived from adult cohorts [42,43,44]. Thus, from the surveys, we excluded 22 subjects (<18 years old). Additionally, we excluded 94 subjects who did not go to the park regularly (i.e., those only going to the park on weekends and for less than three times per week), because even though we acknowledge the possible health benefits of practicing physical activity on weekends, the epidemiological evidence for long-term benefits of physical activity is more robust for those regularly practicing physical activity. Finally, we excluded 196 subjects whose activity in the park could not be classified in a physical activity category according to the physical activity classification described by Ainsworth et al., 2011 [45]. Appendix A shows the number of subjects excluded from the initial sample of survey respondents; and the final number of survey respondents included for our analysis (*N* = 661) (Appendix A).

We estimated the daily number of users visiting the park using both survey data and data from the counting campaign. The study population was classified into distinct groups according to the main activity they conducted in the park: walking for leisure, walking commuters, cyclists, and runners. We subsequently divided each of these groups by age (18 to 64 years old, and ≥65 years old), with the aim of assigning appropriate age-specific incidence rates and exposure-response functions [42,43,44] (Appendix A). We ended excluding runners and walking commuters ≥65 years old due to the low response rate (*N* = 21 runners, and *N* = 5 walking commuters) obtained for this group of users [41]. Energy expenditures associated with the four different types of physical activity were defined in terms of the metabolic equivalent of task (MET) [45].

Other input data sources used in this study to model the health impacts were: (1) The Barcelona Health Survey, used to characterize the base levels of physical activity throughout the Barcelona population [46,47]; (2) epidemiological studies (meta-analysis and prospective cohort studies) to obtain the exposure-response functions between physical activity and a variety of health outcomes [42,43,44]; and (3) Barcelona health records to characterize the age- and sex-specific mortality and incidence rates [46,48,49] (Figure 2).

### 2.2. Study Design and the “Blue Active Tool”

We designed and employed a bespoke quantitative spreadsheet model, the “Blue Active Tool”, using Excel 2007 (Microsoft Corp., Albuquerque, NM, USA) (available from the authors upon request). It is a quantitative tool based on a comparative risk assessment approach. The tool executes the risk characterization of the comparative risk assessment, integrating hazard identification, exposure-response function assessment, and exposure assessment, previously performed by the authors. Thus, the “Blue Active Tool” estimates health and health-related economic benefits of physical activity. Input data (i.e., physical activity behavior of the study population) needs to be provided by the authors, and this is complemented with data provided by other epidemiological studies to obtain exposure-response functions between physical activity and health outcomes. The tool needs to be adjusted to the specific study population in terms of mortality and incidence rates, and population exposed. Using the “Blue Active Tool”, we quantified potential health and health-related economic benefits of performing physical activity in the park using two scenarios. Scenario 1 assumed that 100% of the reported physical activity was new since the park regeneration and related directly to the new infrastructure. Scenario 2 was more conservative, assuming that only 50% of the reported physical activity was new and related to the new infrastructure. We define "new physical activity" as an activity that did not exist before the new infrastructure. In other words, this is not a physical activity that was previously done elsewhere (e.g., in a gym or another park) and moved to the new infrastructure due to the urban riverside regeneration.

The “Blue Active Tool” provides estimates of the health impacts in terms of all-cause mortality, morbidity, and disability-adjusted life years (DALYs), as well as health economic assessment in terms of the value of statistical life (VSL) and direct health costs (Figure 2 and Figure 3). The tool estimates the impacts for each type of physical activity and age groups, providing a central estimate with 95% confidence intervals (CIs). The individual parts of the “Blue Active Tool” are described below.

#### 2.2.1. The “Blue Active Tool”: Physical Activity and Health Outcomes Modelling

The “Blue Active Tool” modelled exposure-response between physical activity and all-cause mortality in a non-linear function [44]. For morbidity outcomes, the non-linear exposure-response function was also applied using the same function as for mortality [42,43,44] (Appendix A). It was assumed that the base levels of physical activity of the study population were similar to those reported for the population of Barcelona [46,47], because data on the base levels of physical activity of the specific study population was not available (Appendix A). Levels of the new physical activity performed in the Besòs Riverside Park were estimated in METs, using the park user survey and counting campaign data as previously described. We obtained age- and sex-specific exposure-response of physical activity and all-cause mortality and specific diseases [including ischemic heart disease (IHD), ischemic stroke, type 2 diabetes (DM2), colon cancer, breast cancer, and dementia (Appendix A)] from prior meta-analyses and prospective cohort studies [42,43,44]. These exposure-response functions were employed to calculate the relative risk (RR) and the population attributable fraction (PAF) for each health outcome, stratified by age and sex, for both scenarios. Using this, we estimated the annual prevented deaths and cases of disease by age and sex [50,51,52]. The analysis was based on age- and sex-specific all-cause mortality and incidence rates derived from the Barcelona population [41,47,48,49]. Health results were also translated into DALYs using a standard approach [53,54]. We multiplied the age- and sex-specific attributable fraction to the corresponding DALYs estimation from Spain, scaled to the study population size, from the Global Burden of Disease Project [55] (Figure 2).

#### 2.2.2. The “Blue Active Tool”: Health Economic Assessment

The health economic assessment was conducted for all-cause mortality (based on the VSL), and for all morbidity outcomes (based on direct health-care costs). We estimated the monetary value of mortality multiplying the VSL for Spain (3,202,968 Euros) [56] by the expected cases of death avoided for each type of physical activity. Direct health costs (i.e., morbidity costs) were estimated multiplying by the expected sex- and age-specific cases of diseases and the direct health-care costs reported for each morbidity outcome in Spain [57,58] (Appendix A). The tool also reports total economic values based on summing the monetary value of mortality and direct health costs.

## 3. Results

### 3.1. Characteristics of the Study Population

It was estimated that the Besòs Riverside Park attracted 5753 adult users per day, engaging in one of the four physical activities included in the analysis (i.e., cycling, running, walking for leisure or walking for commuting) (Table 1). The mean age of our sample was 48 years old, ranging from 18 to 85 years old, with more male than female users (65% vs. 35%, respectively). We estimated that 49% of the users cycled as the main activity conducted in the park; 38% of the users walked for leisure; 12% were runners; and 1% were walking commuters (Appendix A). According to the surveys, the majority of the users came from towns and cities located next to the Besòs River [41].

### 3.2. Health and Health-Related Economic Impacts

Among the 5753 users, in Scenario 1, assuming that 100% of the physical activity conducted in the Park was new and occurred due to the park regeneration intervention, we estimated an annual reduction of 7.3 deaths (95% CI: 5.4; 10.2), 6.2 cases of different diseases (95% CI: 2.0; 11.6), and 11.1 DALYs (95% CI: 3.4; 20.5) (Table 2). Among morbidity outcomes, dementia had the greatest number of cases avoided, with 1.1 annual cases for women (95% CI: 0.4; 2.1) and 3.5 (95% CI: 1.4; 6.3) for men. In terms of annual DALYs, the greatest benefit was also for dementia [3.5 DALYs avoided for men (95% CI: 1.4; 6.4)], followed by IHD [1.8 (95% CI: 0.6; 3.1) and 3.3 (95% CI: 1.2; 5.7) DALYs avoided for women and men, respectively] (Table 2). In Scenario 2, assuming that only 50% of the physical activity conducted in the park was new and due to the intervention, this would result in an annual reduction of 4.8 deaths (95% CI: 3.6; 6.7), 4.1 cases of different diseases (95% CI: 1.0; 7.6), and 7.4 DALYs (95% CI: 1.9; 13.5) (Table 2). In terms of type of activity, in both scenarios, the largest benefit was found for those cycling in the park [e.g., 7.9 DALYs avoided per year (95% CI: 2.4; 14.6) in Scenario 1] (Table 3).

Benefits to population health were converted into estimates of health-related economic benefits. Our estimate of reduced mortality in one year would correspond to a reduction of 23,403,186 euros (95% CI: 14,148,033; 32,787,354) for Scenario 1, and 15,524,195 euros (95% CI: 11,414,915; 21,541,777) for Scenario 2 (Table 2). In terms of direct health-care costs, we estimated an annual reduction of 29,934 euros (95% CI: 10,748; 55,278) for Scenario 1, and 19,849 euros (95% CI: 5171; 36,085) for Scenario 2 (Table 2). The total health-related economic benefits due to the intervention would then be 23,433,120 euros (95% CI: 17,158,781; 32,842,631) per year for Scenario 1, and 15,544,044 euros (95% CI: 11,420,085; 21,577,862) per year for Scenario 2 (Table 2). Cycling, followed by walking for leisure, had the greatest health-related economic impacts in both scenarios (e.g., 84.5% and 15% of the health-related economic impact, respectively in Scenario 1). Only 0.6% of the economic impact was related to running and walking for commuting in both scenarios (Table 3).

### 3.3. Sensitivity Analysis

We conducted a sensitivity analysis considering the minimum visit duration to the park reported by walking commuters (i.e., 30 min/day), instead of the mean visit duration reported by this group of users (Table 1), resulting in a minimum change in the overall results (Appendix A) in both scenarios. This was done because the mean duration reported by walking commuters was longer (98 min/day) than that reported by other user groups (between 58 and 65 min/day, Table 1) [41]. We also performed a sensitivity analysis for those cyclists older than 65 years old, considering that the mean visit duration to the park was 30 min/day, compared to the mean duration reported by this group of users (65 min/day, Table 1). In this case, we still observed health and health-related economic benefits (Appendix A), although these were lower than the benefits observed in the main analysis, both for Scenario 1 and Scenario 2 (Table 3).

## 4. Discussion

### 4.1. Principal Findings

The development of the Besòs Riverside Park in Barcelona was primarily undertaken to improve the ecology of the area, but our assessment demonstrated that this intervention provides health benefits to the population using this infrastructure, by encouraging physical activity. We developed and applied the “Blue Active Tool” to estimate health and health-related economic benefits associated with this physical activity. The results estimated a potential annual health benefit of 11.1 DALYs (95% CI: 3.4; 20.5) among park users. These health benefits were translated into a health-related economic cost reduction of 23.4 million euros per year (95% CI: 17.2; 32.8). The largest health and health-related economic benefits were mainly due to the number of users cycling and walking for leisure (Appendix A). The health and health-related economic benefits were mainly driven by mortality rather than morbidity, similar to those reported by previous studies [51,59].

Previous studies have examined the impacts on health of other types of urban regeneration projects: urban regeneration programs in deprived Dutch districts [31,35] and in Northern Ireland [36]; urban regeneration implying neighborhood demolition and relocation [32]; the regeneration of a port area in a deteriorated region of the Bay of Pasaia—Spain [33]; a vacant lot greening program in Philadelphia U.S. [37]; and the regeneration of a street in the historical centre of Seville—Spain [34]. Results are mixed, with some projects showing positive relationships to health outcomes [31,37]; some reporting little or no benefits [35,36], and others finding inconsistent results [32,33,34]. However, to our knowledge, this is the first study assessing health and health-related economic impacts of an urban riverside park regeneration project.

This study also contributes to the growing evidence on health benefits of both green and blue spaces, given that the Besòs Riverside Park is a combination of both types of natural spaces, which may reinforce the benefits from the two types of natural environments. Our study also shows the potential importance and the impact of urban planning on public health at the city scale. The regeneration of natural environments in urban settings is highly relevant given rapid urbanization globally, and the potentially negative health and well-being impacts of living in cities.

### 4.2. Strengths and Limitations

The aim of this study was to assess health and health-related economic impacts of the physical activity performed on the renovated banks of an urban river. We found health benefits related to physical activity (Table 2 and Table 3), although we only included adults who were regular users, and who reported one of the four main activities (cycling, walking to work or for leisure and running) (Appendix A). Even larger benefits could be expected if all users—including those of other age groups (e.g., children), less frequent users, and users doing other types of physical activity—had been included in the analysis.

An important advantage of the current analysis is that the “Blue Active Tool” modelled the relationship between physical activity and the health outcomes with a non-linear function, providing more conservative estimations of the health benefits compared to using a linear relationship. The tool took into account the base levels of physical activity of the study population (based on Barcelona population data), assuming that health benefits would be distributed according to the base physical activity levels, and acknowledging that more health benefits will be expected in those populations that were originally more sedentary and fewer benefits in those that were already more active prior to the intervention. Due to the lack of available data specific on physical activity levels from those living in the surroundings of the riverside park, we assumed that physical activity levels of the study population were similar to the Barcelona population, despite potential differences between socioeconomic characteristics (Appendix A). In addition, this study also captured the possible seasonal variability on outdoor physical activity practice, considering user surveys, with data from three different months of the year.

Although multiple health outcomes have been related to physical activity, the “Blue Active Tool” only estimates the health impacts of those outcomes with available exposure-response functions from previous meta-analyses (i.e., all-cause mortality, IHD, ischemic stroke, DM2, colon and breast cancer, and dementia) (Figure 3) [42,43,44]. In addition, this study only focused on physical activity, although other health determinants could be related to the Urban Riverside Park as well (Figure 3). For example, the promotion of social cohesion or social interaction (which in turn have impacts on mental health and well-being); or the attenuation of noise, air pollution, and extreme temperatures—ecosystem services (i.e., direct and indirect contributions of ecosystems to health and well-being) which were not considered within the scope of this study—Figure 3. Besides this, the exposure-response functions employed by the “Blue Active Tool” were obtained from other epidemiological studies, which already considered other covariates [42,43,44].

Besides the health benefits associated with physical activity, the risks associated with the use of urban parks such as bicycle accidents, a runner having a heart attack, sunburn, sunstroke, pollen allergies, air pollution exposure, safety concern (rape, robbery, assault…), etc., should be also considered. However, due to the lack of data to estimate these risks, we have not included them in the assessment (Figure 3). Nevertheless, previous studies have reported that physical activity benefits could outweigh the risks related to—for example—air pollution or traffic accident exposures [60,61,62].

Another limitation was the necessity to make assumptions (summarised in Appendix A). Acknowledging that there might be some displacement of physical activity from spaces existing before the urban riverside park (e.g., gyms, parks, beach, etc.), we designed two scenarios assuming different proportion of new physical activity performed in the riverside Park. For Scenario 1, we assumed that 100% of the physical activity performed in the park was new. In Scenario 2, we assumed that only 50% of the physical activity performed in the park was new. We created these scenarios because of the lack of specific data on the user physical activity behavior before the intervention. Of note, a previous study in Barcelona on urban cyclists [63] suggested that physical activity related to bicycle commuting performed using new bicycle infrastructure represented an additional physical activity, rather than a substitution of prior regular physical activity. This extra physical activity was the result of performing more moderate physical activity while travelling by bicycle, showing a positive dose-response relationship between bicycle commuting and physical activity duration. Moreover, physical activity practiced on the riverbanks of the Besòs River after the urban riverside regeneration might bring more health and well-being benefits than physical activity practiced in grey urban settings or indoors because it is being practiced in a natural environment, in green and blue spaces, where we expect to find lower levels of air pollution, temperature, and noise [18,64,65]. Another assumption that we made in this study was to consider the sample of survey respondents as representative of the park users (Appendix A). Even though the surveys were conducted by the local authorities of Barcelona, a clear description of the methods used to recruit the participants for this survey was not available. Thus, the procedure employed to collect these data could include a potential selection bias, which might have affected the representativeness of the sample of this study.

Finally, although not of direct relevance to the current analysis, gentrification could be a negative long-term consequence of this urban riverside regeneration. Gentrification has been defined as the displacement of people from one neighborhood to another as a result of increased costs in the restored area (e.g., higher rents) [66,67]. Over time, the creation of the Besòs Riverside Park could impact local property values and increase the affluence of nearby neighborhoods. In turn, this could change the type of neighborhood amenities and services available, leading to an increased cost of living in the area, and stimulating the real estate speculation, resulting in health inequalities due to the displacement of the poorer residents [68]. In this case, residents forced to move out due to economic reasons would not benefit from the health effects estimated in this study. However, gentrification has no presumably occurred in the case of the Besòs Riverside Park, given that the pattern of the average rental price (€/month) in Sant Adrià de Besòs and Santa Coloma de Gramenet from 2005 to 2015 was similar than for Barcelona and other municipalities of the metropolitan area (data not shown) [69]. The implications of this possible gentrification effects were not included in this analysis, because it goes beyond the scope of this study. However, we acknowledge the importance of gentrification; and for this reason, we suggest that all urban regeneration projects should be accompanied with policies and regulations to impede or reduce the gentrification effects on existing inhabitants (e.g., safeguard affordable housing, protect senior homeowners, land use regulation, etc.).

### 4.3. Implications and Recommendations

The implementation of urban riverside regenerations, similar to the one evaluated in the present study, should be expanded in cities to promote the practice of physical activity among the population. As suggested in this study, such interventions might bring health and health-related economic benefits to the population. It is also important to improve the existing green or blue infrastructures by facilitating the accessibility, the aesthetics, and providing good maintenance to sustain and even increase their usability by attracting more users to these natural environments that already exist in the urban areas.

Currently there is a lack of evidence on the health implications of regeneration of urban natural spaces, so more research is needed in this area. More evidence on this area will help policy makers and stakeholders to improve urban planning, creating healthy urban environments and promoting health in all policies’ approach. However, in order to create this scientific evidence, it will be necessary to have data available and accessible to characterize and define urban interventions, populations, user behaviors, and local health data. Moreover, since urban environmental interventions may benefit more socio-economic deprived populations [70], further research should focus on the assessment of health inequities in these groups. The design and development of these urban interventions must guarantee the equal use and enjoyment among all the population considering different age groups, gender, ethnicity, or socioeconomic status.

For the specific case of the Besòs Riverside Park, the incorporation of trees (that would create shade) along the riverbanks, and campaigns promoting different activities for all ages, might be initiatives that could increase the usability of the park between the citizens. Furthermore, investments in the increase of natural public spaces (both blue and green spaces) in other parts of the city will also help to promote health and well-being across the city population.

## 5. Conclusions

The number of people living in urban areas worldwide is increasing. Thus, nature-based urban planning solutions, such as urban riverside regeneration, should be considered as a relevant contributor to improving urban health and well-being, especially via the mechanism of increased physical activity.

## Figures and Tables

**Figure 1 ijerph-16-00462-f001:**
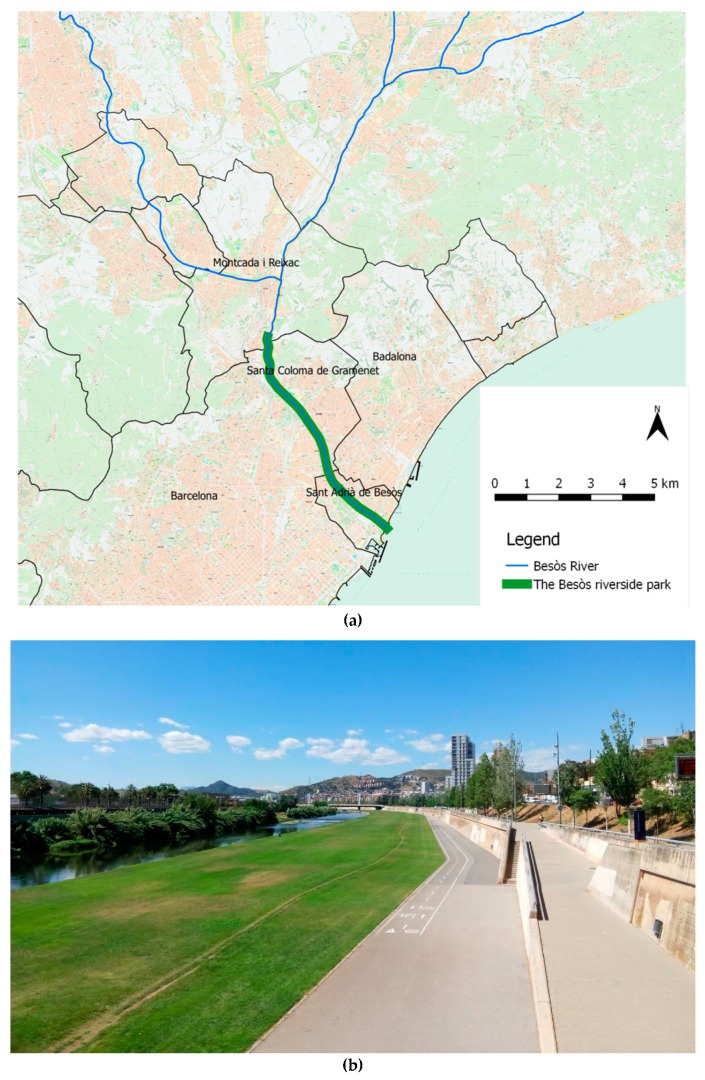
Setting of the study area: (**a**) Map of the Besòs Riverside Park, which spans the last 9 km of the Besòs River [40]; (**b**) The Besòs Riverside Park (Image: Cristina Vert/ISGlobal, May 2017).

**Figure 2 ijerph-16-00462-f002:**
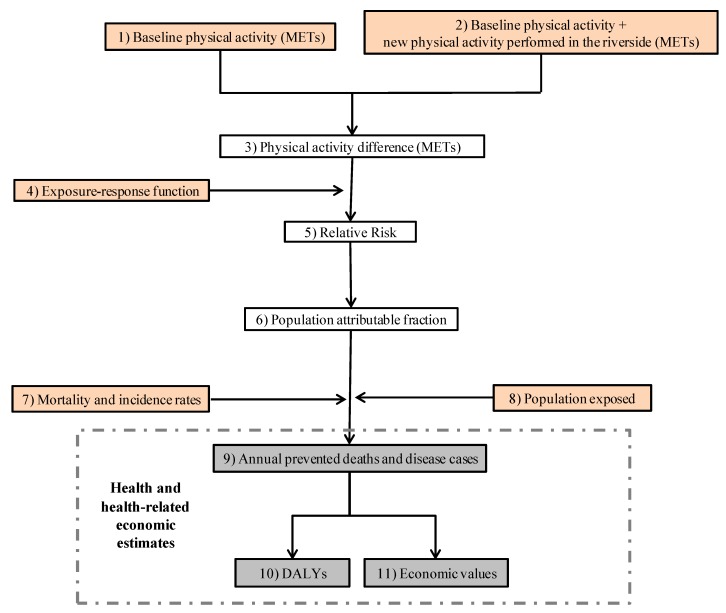
Methodological approach of the “Blue Active Tool”. In orange, the input data used in this study to estimate health and health-related economic impacts of the physical activity related to the urban riverside park. METs: Metabolic equivalent of task; DALYs: Disability-adjusted life years.

**Figure 3 ijerph-16-00462-f003:**
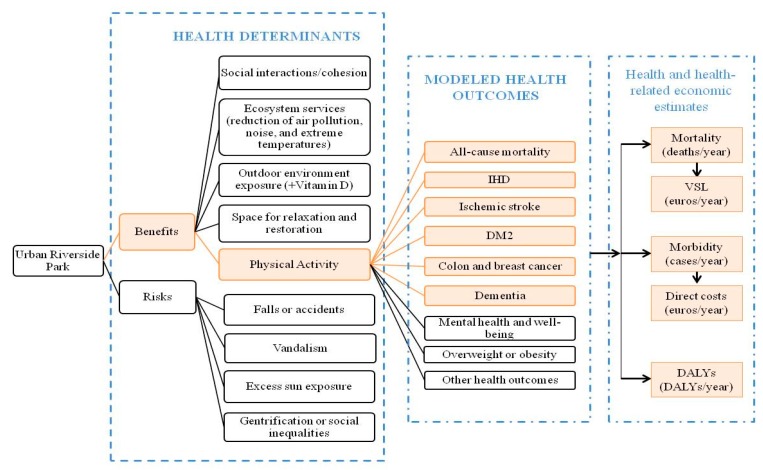
Pathways modeled (in orange) and non-modeled (in black) related to health impacts derived of the urban riverside park. IHD: ischemic heart disease; DM2: diabetes mellitus type 2; VSL: value of statistical life; DALYs: Disability Adjusted Life Years.

**Table 1 ijerph-16-00462-t001:** Input data (for Scenario 1 and Scenario 2) on adult users of the park and key assumptions used in the model.

Types of Physical Activity	METs per Type of Physical Activity ^a^	Mean Duration of Visits to the Park ^b^ (min/Day)	Mean Frequency of Visits to the Park ^c^ (Days/Week)		Estimated Visits/Day ^d^	Estimated METs Hour/Week per Subject ^e^		Estimated Park Users/Day ^f^ (*N*)
			Scenario 1	Scenario 2		Scenario 1	Scenario 2	
Walking for leisure								
≥18 and ≤64 years old	3.5	59	5	2.5	1	17	9	1566
≥65 years old	3.5	63	5	2.5	1	18	9	619
Cycling								
≥18 and ≤64 years old	7.5	65	5	2.5	1	41	20	535
≥65 years old	7.5	65	5	2.5	1	41	20	2287
Running								
≥18 and ≤64 years old	7.0	58	5	2.5	1	34	17	686
Walking for commuting								
≥18 and ≤64 years old	4.0	98	5	2.5	2	65	33	60
Total (all users)								5753
≥18 and ≤64 years old								2848
≥65 years old								2905

^a^ METs = Metabolic Equivalent of Task. These values have been assigned for each type of physical activity according to Ainsworth et al. 2011 [45]. ^b,c^ Mean duration and frequency of user’s visits to the park, based on the data provided by the surveys [41] (Appendix A) for Scenario 1. For Scenario 2, we have considered the 50% of the frequency values reported. ^d^ This information was not provided by the surveys. Thus, we assumed that in Scenario 1 subjects would visit the Park once a day. Except the walking commuters, who need to go to and from work. Thus, we assumed that in Scenario 1, this group of users would visit the park twice a day (Appendix A). ^e^ Value obtained by multiplying the input data [(METs) × (mean duration) × (mean frequency) × (number of visits/day)]. ^f^ The number of “park users/day” was estimated by using the total users from counting campaign, and the proportion of users by type of activity from the surveys.

**Table 2 ijerph-16-00462-t002:** Results in annual cases of mortality, diseases, DALYs, and health-related economic outcomes due to the intervention. Results provided for Scenario 1 (assuming that 100% of the physical activity conducted in the park was new physical activity), and Scenario 2 (assuming that 50% of the physical activity conducted in the park was new physical activity).

Health Outcomes	Scenario 1			Scenario 2		
	Cases/Year (95% CI)	DALYs/Year (95% CI)	Euros/Year (95% CI)	Cases/Year (95% CI)	DALYs/Year (95% CI)	Euros/Year (95% CI)
All-cause mortality	−7.3 (−10.2, −5.4)	−	−23,403,186 (−32,787,354, −17,148,033)	−4.8 (−6.7, −3.6)	−	−15,524,195 (−21,541,777, −11,414,915)
Diseases						
IHD (W)	−0.1 (−0.1, 0.0)	−1.8 (−3.1, −0.6)	−61 (−105, −22)	0.0 (−0.1, 0.0)	−1.2 (−2.1, −0.4)	−41 (−70, −15)
IHD (M)	−0.4 (−0.6, −0.1)	−3.3 (−5.7, −1.2)	−421 (−727, −151)	−0.3 (−0.4, −0.1)	−2.2 (−3.8, −0.8)	−282 (−485, −102)
Stroke (W)	−0.1 (−0.3, 0.0)	−0.1 (−0.3, 0.0)	−271 (−590, 0)	−0.1 (−0.2, 0.0)	−0.1 (−0.2, 0.0)	−182 (−393, 0)
Stroke (M)	−0.8 (−1.8, 0.0)	−0.5 (−1.1, 0.0)	−1790 (−3903, 0)	−0.5 (−1.2, 0.0)	−0.4 (−0.8, 0.0)	−1206 (−2601, 0)
DM2 (W)	−0.1 (−0.1, 0.0)	−0.1 (−0.2, 0.0)	−199 (−336, −39)	0.0 (−0.1, 0.0)	−0.1 (−0.2, 0.0)	−135 (−227, −26)
DM2 (M)	−0.1 (−0.2, 0.0)	−0.3 (−0.4, −0.1)	−365 (−615, −71)	−0.1 (−0.1, 0.0)	−0.2 (−0.3, 0.0)	−247 (−416, −48)
Colon C (W)	0.0 (0.0, 0.0)	−0.1 (−0.4, 0.1)	−21 (−60, 14)	0.0 (0.0, 0.0)	−0.1 (−0.3, 0.1)	−14 (−41, 10)
Colon C (M)	0.0 (−0.1, 0.0)	−0.3 (−0.8, 0.2)	−70 (−204, 48)	0.0 (0.0, 0.0)	−0.2 (−0.5, 0.1)	−47 (−137, 33)
Breast C (W)	0.0 (0.0, 0.0)	0.0 (−0.1, 0.0)	−11 (−25, 2)	0.0 (0.0, 0.0)	0.0 (−0.1, 0.0)	−7 (−17, 2)
Dementia (W)	−1.1 (−2.1, −0.4)	−1.0 (−1.8, −0.4)	−6573 (−11,980, −2589)	−0.7 (−1.3, −0.2)	−0.7 (−1.2, −0.2)	−4350 (−7795, −1235)
Dementia (M)	−3.5 (−6.3, −1.4)	−3.5 (−6.4, −1.4)	−20,154 (−36,733, −940)	−2.3 (−4.1, −0.6)	−2.3 (−4.2, −0.7)	−13,337 (−23,903, −3788)
All diseases	−6.2 (−11.6, −2.0)	−11.1 (−20.5, −3.4)	−29,934 (−55,278, −10,748)	−4.1 (−7.6, −1.0)	−7.4 (−13.5, −1.9)	−19,849 (36,085, −5171)
Total (euros/year)			−23,433,120 (−32,842,631, −17,158,781)			−15,544,044 (−21,577,862, −11,420,085)

DALYs: Disability Adjusted Life Years; IHD: ischemic heart disease; DM2: diabetes mellitus type 2; M: men; W: women.

**Table 3 ijerph-16-00462-t003:** Results by type of physical activity, in annual DALY, direct health-care costs, and value of statistical life (Scenario 1 and Scenario 2).

Types of Physical Activity	Scenario 1			Scenario 2		
	DALYs/Year (95% CI)	Direct Costs (Euros/Year) (95% CI)	VSL (Euros/Year) (95% CI)	DALYs/Year (95% CI)	Direct Costs (Euros/Year) (95% CI)	VSL (Euros/Year) (95% CI)
Cycling	−7.9 (−14.6, −2.4)	−25,284 (−46,826, −9108)	−15,629,701 (−21,916,593, −11,401,939)	−5.3 (−9.7, −1.3)	−16,818 (−30,648, −4090)	−10,426,408 (−14,505,355, −7,651,506)
Walking for leisure	−2.4 (−4.3, −0.7)	−4487 (−8154, −1608)	−7,255,016 (−10,144,657, −5,367,509)	−1.6 (−2.8, −0.5)	−2920 (−5236, −1059)	−4,753,055 (−6,557,344, −3,510,218)
Running	−0.8 (−1.4, −0.2)	−146 (−264, −28)	−460,256 (−643,971, −336,315)	−0.5 (−0.9, −0.1)	−99 (−178, −19)	−305,284 (−423,907, −224,357)
Walking to work	−0.1 (−0.2, 0.0)	−18 (−33, −3)	−58,213 (−82,133, −42,271)	−0.1 (−0.1, 0.0)	−13 (−23, −2)	−39,448 (−55,172, −28,834)
TOTAL	−11.1 (−20.5, −3.4)	−29,934 (−55,278, −10,748)	−23,403,186 (−32,787,354, −17,148,033)	−7.4 (−13.5, −1.9)	−19,849 (−36,085, −5171)	−15,524,195 (−21,541,777, −11,414,915)

DALYs: Disability Adjusted Life Years; VSL: value of statistical life.

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
