# Peer review of "Health Benefits of Physical Activity Related to an Urban Riverside Regeneration"

_ijerph, 2019, doi:10.3390/ijerph16030462_

Reviewer 1 Report

Thank you for the opportunity to review your work.  Please see the suggested comments.

Overall comments:

It would be better to integrate your supplementary material into manuscript. Due to separated texts and tables, a reader could be easily getting distracted.

2.2.1

Since your study heavily depends on the Blue Active Tool and other determinants (e.g., stressors, other health behaviors such as dietary practices or health screening, social interaction and social support- regarding this, please refer “Social determinants of health and environmental health promotion: Northridge, Sclar, & Biswas, 2003”) might make an impact on health outcome, it is necessary to explain or summary how the Blue Active Tool controls other variables to affect health outcome.

Author Response

ANSWER TO REVIEWERS’ COMMENTS

*Please, find attached a PDF file with the answer to reviwers' comments. I also copy it below. 

Reviewer 1

Thank you for the opportunity to review your work. Please see the suggested comments.

- We thank the reviewer for his/her comments and his/her contribution by reviewing the manuscript. Below, please find an itemized response to all the comments provided. The changes are highlighted in red in the revised manuscript.

Overall comments:

It would be better to integrate your supplementary material into manuscript. Due to separated texts and tables, a reader could be easily getting distracted.

- We acknowledge the reviewer’s suggestion. We have ensured that the main informative tables and figures are in the manuscript. The remaining tables and figures remain in the Supplementary Material because the content of the manuscript can be understood without these tables and figures and/or because the data provided in these tables and figures in the Supplementary Material is clearly referenced in the manuscript. For this reasons we decided to maintain the tables and figures in the Supplementary Material.

2.2.1

Since your study heavily depends on the Blue Active Tool and other determinants (e.g., stressors, other health behaviors such as dietary practices or health screening, social interaction and social support regarding this, please refer “Social determinants of health and environmental health promotion: Northridge, Sclar, & Biswas, 2003”) might make an impact on health outcome, it is necessary to explain or summary how the Blue Active Tool controls other variables to affect health outcome.

- We thank the reviewer for this comment. We agree with the reviewer that other determinants might impact on the health outcomes included in this study. However, our study focused on the assessment of the effects of physical activity. The assessment of other health determinants (beside physical activity) was not part of our study, although we acknowledge their relevance. This is shown in Figure 3 (the pathways modelled in this study are coloured in orange), where other possible determinants are mentioned, although it is clarified that this study focused only on physical activity. In Figure 3 we have only included some health determinants we thought population might be exposed to due to the Urban Riverside Park. Thus, we have not included other health determinants like “dietary practices” or “health screening” as suggested by the reviewer.

Besides this, the Blue Active Tool uses exposure-response functions between physical activity and different health outcomes obtained from other epidemiological studies in which other covariates, such as vegetable intake, social support or alcohol consumption, were already considered. To further clarify it, we have amended the text in section “4.2. Strengths and Limitations”:

“Also, this study only focused on physical activity, although other health determinants could be related to the Urban Riverside Park as well (Figure 3). For example the promotion of social cohesion or social interaction (which in turn have impacts on mental health and well-being); or the attenuation of noise, air pollution, and extreme temperatures – ecosystem services (i.e. direct and indirect contributions of ecosystems to health and well-being) which were not considered within the scope of this study – (Figure 3). Besides this, the exposure-response functions employed by the “Blue Active Tool”, were obtained from other epidemiological studies which already considered other covariates [42,43,47]”.

Reviewer 2 Report

General comments

The authors have addressed an important and relevant topic for environmental public health. By using data from different sources, the authors estimated the health and economic (health-related) impact of an urban riverside regeneration in Barcelona. The paper is easy-to-read, the methods are correct, and the presentation of the results is clear. However, I think that, before a final publication, the authors will need to reconsider some aspects. I will organize my review through the different sections of the paper.

Specific comments

Introduction

The authors set the problem and the relevance of the topic, and the use of references is correct. In this section some small suggestions that I think will help the future reader of this article:

Reference [20] addresses mainly the comparison between outdoor and indoor physical activity, not specifically about the difference between physical activity in built and natural environment. I would suggest dropping it as you have enough justification with the other references.

Figure 1. I would suggest the author to modify this figure. In panel (a) it is not clear in which part of Barcelona the riverside regeneration took place. I would suggest a map showing the whole metropolitan area of Barcelona, and then framing separately the area of the riverside.

Methods

The methods are clearly described. Particularly, figures 2 and 3 help the reader while understanding all data sources and assumptions. Some aspects that should be considered:

P4L8 Is the data of the surveys publicly available? If so, where?

P4L18 I would suggest to include a brief description of types of physical activity described by Ainsworth et al.

P4L22 I won’t use a reference to Table 1 in the methods before describing briefly both scenarios, as I found it confusing.

P6 There’s an error that the page numbering (in page 6, it starts again with page 1). It happens the same in other parts of the text.

P6L12 Can you provide a longer description of the “Blue active tool”? If difficult due to space, can you provide some reference to other study or a description of the tool?

P7L6 I find problematic the assumption that physical activity of the people using the park is similar to the estimates for Barcelona. I would suggest the authors to include a sensitivity analysis assuming baseline physical activity estimates that are closer to the population living near the riverside, at least including SES variability. Also, could be that, within people living in that area, only those with greater levels of physical activity are those using the park. Despite this is addressed partially in the discussion, an analytical strategy to tackle this could be needed. These are my greatest concerns of the study.

Results

 Results are clearly described. Tables are well referenced, and all results answer to the objectives of the study. Just one small comment on P12L11; I’ll move the sentence that starts with “These results[…]” to the discussion, as it is an explanation of your results.

Discussion

P14L7 I would say that gentrification is part of your analysis and could be a potential limitation. If people living alongside the area is displaced by wealthier people (which will be more active due to the physical activity SES gradient), then all the benefits that you’ve seen are not for the people living alongside the river. Also, as your assumption is that physical activity of the people using the park is similar to the population of Barcelona, if gentrification has happened alongside the river your results could be biased as people using the park could be more active than Barcelona’s mean.  

Conclusions

Your conclusions only include a mention to the implications. I will include specific numbers of your estimations in the conclusion.

Author Response

ANSWER TO REVIEWERS’ COMMENTS

*Please, find attached a PDF file with all the answer to the reviewers' comments. I've also copied it below. 

Reviewer 2

General comments

The authors have addressed an important and relevant topic for environmental public health. By using data from different sources, the authors estimated the health and economic (health-related) impact of an urban riverside regeneration in Barcelona. The paper is easy to-read, the methods are correct, and the presentation of the results is clear. However, I think that, before a final publication, the authors will need to reconsider some aspects. I will organize my review through the different sections of the paper.

-          We really appreciate reviewer’s words and his/her contribution by reviewing the manuscript. Please, find below an itemized response to all the comments provided. The changes are highlighted in red in the revised manuscript.

Specific comments

Introduction

The authors set the problem and the relevance of the topic, and the use of references is correct. In this section some small suggestions that I think will help the future reader of this article:

Reference [20] addresses mainly the comparison between outdoor and indoor physical activity, not specifically about the difference between physical activity in built and natural environment. I would suggest dropping it as you have enough justification with the other references.

-          We have deleted this reference.

Figure 1. I would suggest the author to modify this figure. In panel (a) it is not clear in which part of Barcelona the riverside regeneration took place. I would suggest a map showing the whole metropolitan area of Barcelona, and then framing separately the area of the riverside.

-          We thank the reviewer for bringing this possible confusion to our attention. We have modified Figure 1 to clearly show the area of study (i.e. Besòs Riverside Park).

*Please, see the image in the PDF attached.

Methods

The methods are clearly described. Particularly, figures 2 and 3 help the reader while understanding all data sources and assumptions. Some aspects that should be considered:

P4L8 Is the data of the surveys publicly available? If so, where?

-          The data of the surveys is not publicly available. We requested this data to “Consorci Besòs”, a Barcelona local authority which was the responsible of the survey data collection. We signed a confidentiality document authorizing the use of the data. However, we have included a reference of the data source.

P4L18 I would suggest to include a brief description of types of physical activity described by Ainsworth et al.

-          We thank the reviewer for this comment. According to the reviewer’s suggestion, we have added the following paragraph on the footnote of Figure S1 (Supplementary Material):

“Users who responded “to be healthy” or “others” as the “reason to come to the Besòs River” in the survey (see Supplementary Material – Table S1), were excluded of the sample of this study. This was because these activities could not be classified in a physical activity category according to the physical activity classification described by Ainsworth et al. 2011 [1]. This classification provides the energy cost of a wide variety of defined physical activities (e.g. dancing, walking, cycling, doing home activities like mopping or cleaning windows, etc.), which can be compared with other epidemiological studies providing data on self-reported physical activity”.  

Part of abovementioned information was already provided in the footnote of Table S1 (Supplementary Material). To clarify, in some cases physical activity was not clearly defined in the survey, and could not be identified among any physical activity category described by Ainsworth et al. Thus, we could not include subjects reporting these unspecific types of physical activity. Table 1 shows the physical activity categories included in the current study along with their corresponding energy expenditure in METs.

P4L22 I won’t use a reference to Table 1 in the methods before describing briefly both scenarios, as I found it confusing.

-          We thank the reviewer for bringing this possible confusion to our attention. We have deleted the reference to Table 1 in this part of the methods. And Table 1 has been moved to the “Results” section. See the paragraph which has been modified:

We estimated the daily number of users visiting the Park using both survey data and data from the counting campaign. The study population was classified into distinct groups according to the main activity they conducted in the Park: walking for leisure, walking commuters, cyclists, and runners. We subsequently divided each of these groups by age (18 to 64 years old, and ≥ 65 years old), with the aim of assigning appropriate age-specific incidence rates and exposure-response functions [42,43] (Supplementary Material – Table S2). We ended excluding runners and walking commuters ≥ 65 years old due to the low response rate (N=21 runners, and N=5 walking commuters) obtained for this group of users [41]. Energy expenditures associated with the four different types of physical activity were defined in terms of the metabolic equivalent of task (MET) [44]”.    

P6 There’s an error that the page numbering (in page 6, it starts again with page 1). It happens the same in other parts of the text.

-          Thank you for your comment. The reviewer is right. We did not include page numbers in the original manuscript we submitted. Thus, this might be done by the editorial office for edit formatting. We prefer not to modify it to not interfere with editorial office work.  

P6L12 Can you provide a longer description of the “Blue active tool”? If difficult due to space, can you provide some reference to other study or a description of the tool?

-          According to the reviewer’s suggestion, we have provided a description of the “Blue Active Tool” at the beginning of section 2.2. Study Design and the “Blue Active Tool””:

“We designed and employed a bespoke quantitative spreadsheet model, the “Blue Active Tool,” using Excel 2007 (Microsoft) (available from the authors upon request). It is a quantitative tool based on a comparative risk assessment approach. The tool executes the risk characterization of the comparative risk assessment, integrating the hazard identification, exposure-response function assessment, and exposure assessment, previously performed by the authors. So, the “Blue Active Tool” estimates health and health-related economic benefits of doing physical activity. Input data (i.e. physical activity behavior of the study population) needs to be provided by the authors, and this is complemented with data provided by other epidemiological studies to obtain exposure-response functions between physical activity and health outcomes. The tool needs to be adjusted to the specific study population in terms of mortality and incidence rates, and population exposed. Using the “Blue Active Tool”, we quantified potential health and health-related economic benefits of performing physical activity in the Park using two Scenarios. […]”.

P7L6 I find problematic the assumption that physical activity of the people using the park is similar to the estimates for Barcelona. I would suggest the authors to include a sensitivity analysis assuming baseline physical activity estimates that are closer to the population living near the riverside, at least including SES variability. Also, could be that, within people living in that area, only those with greater levels of physical activity are those using the park. Despite this is addressed partially in the discussion, an analytical strategy to tackle this could be needed. These are my greatest concerns of the study.

-          We thank and acknowledge the reviewer’s concern about assuming that physical activity of the people using the park is similar to the estimates for Barcelona. However, as mentioned in Table S8 of the Supplementary Material, “data on the base levels of physical activity of the specific study population was not available” (i.e. absence of physical activity data by small geographical location or SES). Thus, it is not possible to conduct a sensitivity analysis “assuming baseline physical activity estimates that are closer to the population living near the riverside”, as suggested by the reviewer, because this data is not available. We used the base levels of physical activity for the population of Barcelona since this was the most similar and comparable data available. We also agree with the reviewer that SES is an important determinant of physical activity levels, and the relevance to use SES for a sensitivity analysis, but this data is either available. Regarding the second comment raised by the reviewer, we agree that interventions like the urban riverside regeneration project tend to promote physical activity especially among the population who is already active. However, we assumed that not only the most active people were the main river park users. Moreover, we are aware of medical recommendations that have been applied to the sedentary and/or old population in the surroundings of the Besòs Riverside Park to promote physical activity by using the riverside park as complement of their daily activities. We have now added the following text in the 4.2. Strengths and Limitations” section:

“Due to the lack of available data specific on  physical activity levels from those living in the surroundings of the riverside park, we assumed that physical activity levels of the study population were similar to the Barcelona population, despite potential differences between socioeconomic characteristics (Table S8 – Supplementary Material)”.

We have also amended the text in the “2.2.1. The “Blue Active Tool”: Physical Activity and Health Outcomes Modelling” section:

“It was assumed that the base levels of physical activity of the study population were similar to those reported for the population of Barcelona [45,46] because data on the base levels of physical activity of the specific study population was not available (Supplementary Material – Tables S3 and S8)”.

Results

 Results are clearly described. Tables are well referenced, and all results answer to the objectives of the study. Just one small comment on P12L11; I’ll move the sentence that starts with “These results […]” to the discussion, as it is an explanation of your results.

-          We thank the reviewer for this comment. According to the reviewer’s suggestion, we have merged the abovementioned sentence with a similar one in the “Discussion”, section “4.1. Principal Findings”:

The largest health and health-related economic benefits were mainly due to the number of users cycling and walking for leisure (Supplementary Material – Table S5).”

Discussion

P14L7 I would say that gentrification is part of your analysis and could be a potential limitation. If people living alongside the area is displaced by wealthier people (which will be more active due to the physical activity SES gradient), then all the benefits that you’ve seen are not for the people living alongside the river. Also, as your assumption is that physical activity of the people using the park is similar to the population of Barcelona, if gentrification has happened alongside the river your results could be biased as people using the park could be more active than Barcelona’s mean.

-          We agree with the reviewer and we consider this point has been partially addressed in the last paragraph of section 4.2. Strengths and Limitations”. However, to make it more clear, we have included the following sentence in the same section:

In this case, residents forced to move out due to economic reasons would not benefit from the health effects estimated in this study”.

-          We also agree with the reviewer that in case gentrification happened, our results would be biased because people using the park could be more active (and healthier) than Barcelona’s mean. However, although we think gentrification can be a negative consequence of any urban regeneration project, we do not think gentrification have occurred yet in the case of the Besòs Riverside Park. According to data provided by the Metropolitan Area of Barcelona, the pattern of the average rental price (€/month) in Sant Adrià de Besòs and Santa Coloma de Gramenet from 2005 to 2015, was similar than for Barcelona (please, see the graph below). Thus, the cost of living in the area alongside the Besòs Riverside Park has not increased more than in the city of Barcelona. The raise observed in 2008 was probably due to the economical crisis that affected all the country (like other parts in Europe). We observed the same tendency comparing the average rental price with other municipalities of the Metropolitan Area of Barcelona (data not shown). We have added the following sentence in the “4.2. Strengths and Limitations” section:

However, gentrification has no presumably occurred in the case of the Besòs Riverside Park, given that the pattern of the average rental price (€/month) in Sant Adrià de Besòs and Santa Coloma de Gramenet from 2005 to 2015 was similar than for Barcelona and other municipalities of the metropolitan area (data not shown) [add ref.]*.”

*[ref]: We would really appreciate the following reference to be added in the manuscript at the end of the above written sentence, as indicated above. Thank you. Reference:  Àrea Metropolitana de Barcelona Sistema d’Indicadors Metropolitans de Barcelona Available online:          https://iermbdb.uab.cat/index.php?ap=0&id_ind=1660&id_cat=402.

*Please, see the image in the PDF attached.

Conclusions

Your conclusions only include a mention to the implications. I will include specific numbers of your estimations in the conclusion.

-          We thank the reviewer for this suggestion. However, since we have already included specific numbers of our estimations in the “Results” and “Discussion” sections, we do not think it is also necessary to provide it in the “Conclusions”.

Round  2

Reviewer 2 Report

The authors have addressed most of the issues that were suggested in the first review. When they were not fully covered, the justification is precise.